# Secure Communication Model for Quantum Federated Learning: A Proof of Concept

**Dev Gurung** [†] **, Shiva Raj Pokhrel and Gang Li**
School of Information Technology
Deakin University, VIC, AUS
{d.gurung, shiva.pokhrel, gang.li}@deakin.edu.au

## ABSTRACT

We design a model of Post Quantum Cryptography (PQC) Quantum Federated Learning (QFL). We develop a proof of concept with a dynamic server selection and study convergence and security conditions. We develop a preliminary study with a proof of concept model of post-quantum secure QFL.

## 1 INTRODUCTION

Quantum Federated Learning (QFL) is an emerging area with several studies in recent years; see (Larasati et al.) and references therein. Most QFL works focus on optimization (Kaewpuang et al.; Yang et al.; Yun et al.; Xia & Li), security (Xia et al.; Zhang et al.) and implementation (Qi; Yamany et al.; Abbas et al.; Huang et al.; Chehimi & Saad). However, post-quantum secure QFL is poorly studied in the literature.

We develop a preliminary study with a proof of concept model of post-quantum secure QFL. The two *main contributions* are as follows: i) we develop a novel PQC-QFL model, ii) we implement the proof of concept and evaluate the proposed model theoretically and experimentally. The uploading of learned model parameters, $w_i$, from device $i$ to the server, can be poisoned by an adversary.

Post-quantum secure schemes are essential to protect against unauthorized access during uploads/downloads. We consider a device i $\in$ $\{d_i\}$ with signature scheme $S_i(sk_i, pk_i)$ which is used to sign the parameters $w_i$ as, $\sigma_i \leftarrow S_i^{sign}(w_i, sk_i)$, and are verified as $\{w_i, pk_i, \sigma_i\}$ by the server.

## 2 PROPOSED PQC-QFL MODEL

We present details of the proposed PQC-QFL model in Algorithm 1. For $n$ devices, a server device $d_s$ is selected dynamically (e.g., random server selection). First, all devices train the models locally. Once the local models are learned, each device $d_i$ signs with its private key $sk_i$ to generate the signature $\sigma_i$. Each device sends this information to the server $d_s$ and its public key $pk_i$. Upon receiving this information, the server verifies and filters the compromised models. Eventually, it performs the model aggregation to generate a global model and sends it back to the devices. The PQC-QFL design is grounded in the convergence and security conditions as follows: i) each device trains local Stochastic Gradient Descent (SGD) with $T$ iterations, and PQC scheme with security level $\lambda$, the Algorithm 1 converges at a rate of $n * \{O(1/\sqrt{T}) + O(1/\lambda^2)\}$; ii) with Algorithm 1, the security level is SUF-CMA secure (provided by the Dilithium (Ducas et al.) signature scheme). See the appendix for details.

**Algorithm 1** Post Quantum Secure Communication in QFL

1: Inialize: total $n$ devices in $device\_list$ : $\{d_i\} = \{d_1, d_2, ..., d_n\}$.
2: Generate keys $(pk, sk)$ for each device.
3: Randomly select a server $d_s$ from $\{d_i\}$
4: **procedure** DEVICETASK($pk, sk$)
5:    **for** device $d_i$ in $\{d_i\}$ **do**
6:       Train local params $w_i$.
7:       Sign params $w_i$ with PQC-Scheme: $\sigma_i \leftarrow Sign(w_i)$
8:       Send $\{\sigma_i, w_i, pk_i\}$ to server.
9:    **end for**
10: **end procedure**
11: **procedure** SERVERTASK($pk, sk$)
12:    Initialization: $param\_list = []$
13:    $d_s$ receives $\{\sigma_i, w_i, pk_i\}$ from all $\{d_i\}$.
14:    **for** device $d_i$ in $\{d_i\}$ **do**
15:       $d_s$ verifies each $\sigma_i$.
16:       **if** Verification is true **then**
17:          $param\_list.append(w_i)$
18:       **else**
19:          $w_i$ is excluded
20:       **end if**
21:    **end for**
22:    Perform fedAvg.
23:    Send back global params $w_g$ to devices.
24: **end procedure**

More importantly, with the proposed model, we have demonstrated that the resilience of the proposed dynamic server is always higher. With the proposed PQC-QFL, we have the following two theorems.

**Theorem 1 (Resilience)** *The occurrence of single-point failure with Algorithm 1 (e.g., random-server architecture) is always less than in the current QFL.*

**Theorem 2 (Convergence)** *Given $n$ devices, each device trains local stochastic gradient descent (SGD) with $T$ iterations and employs PQC at security level $\lambda$. Then Algorithm 1 converges with a rate of $n * \{O(1/\sqrt{T}) + O(1/\lambda^2)\}$.*

The proofs are deferred to the appendix.

### 2.1 IMPLEMENTATION AND EVALUATION

We developed our implementation and experimental framework[1] by extending the framework presented by Zhao et al. The post-quantum cryptography, liboqs [2] and liboqs-python [3] libraries are integrated with the codebase. For most experiments, we employ the Dilithium signature scheme. MNIST dataset is shared between clients by following a cycle-m structure where each client will only have a certain number of label classes.

As shown in Figure 1(a), a random server selection performs as demonstrated; PQC-QFL performance is visible after 2 training epochs. The observations are recorded four times per epoch. We observe that with PQC-QFL, quantum machine learning performs very well in training. Figure 1(b), shows the performance on the test set done by the server with the new global model. After the 100 communication round, the test accuracy is around 55%, which is poor compared to classical machine learning. This demonstrates that QFL suffers over non-IID datasets. Figure 1(d) shows the

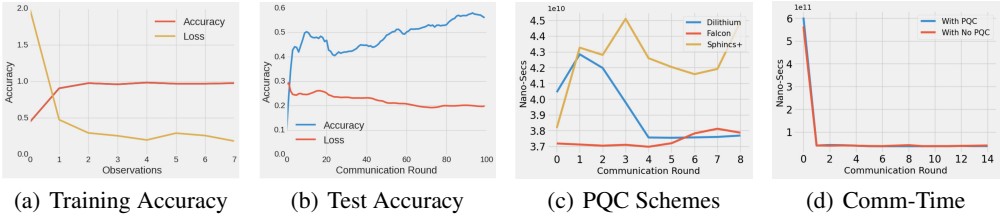

| (a) Training Accuracy | (b) Test Accuracy | (c) PQC Schemes | (d) Comm-Time |

Figure 1: Accuracy, Performance, and Communication time of PQC-QFL.

difference in the impact of using PQC schemes in signing and verification. Some added tasks of the signature scheme would add some delay to the system's overall performance. However, as shown in Figure 1(d), we cannot see much noticeable adverse impact by employing the PQC schemes in terms of nanoseconds. Figure 1(c) shows the comparison of this effect on Dilithium with other schemes such as Falcon (Fouque et al., 2018) and SPINCHS+ (Bernstein et al., 2019). Observe that Dilithium and Falcon perform in a similar fashion. Whereas SPHINCS+ exhibits increased delay compared to Dilithium and Falcon, which requires further investigations and in-depth understanding and is left for future work.

## 3 CONCLUSION

We have developed a novel preliminary model and implementation of PQC-QFL. We performed extensive analysis and presented results that assert the practicality and suitability of PQC schemes in the QFL. The proof of concept experiment also involved the removal of a fixed central server approach where any participating device can be selected to perform as a central server in contrast to a dedicated server in the traditional QFL.

---

[1]https://github.com/s222416822/PQC-QFL-Model

[2]https://github.com/open-quantum-safe/liboqs.git

[3]https://github.com/open-quantum-safe/liboqs-python.git

## URM STATEMENT

Author[†] meets the URM criteria of ICLR 2023 Tiny Papers Track.

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

## APPENDIX: THEORETICAL ANALYSIS

**Theorem 1**. *The occurrence of single-point failure with Algorithm 1 (e.g., random-server architecture) is always less than in current QFL.*

PROOF. With Algorithm 1, the server role is selected randomly or based on the reputation as required. Thus, the server keeps changing with time. For an adversary $A$, to attack fixed-server $fixedServer$, as identity is the same every time, the probability of adversary accuracy attacking the server is $1$. However, with a random or server selection approach, the probability of an adversary $A$ finding the right device $randomServer$ to attack is almost always $< 1$. Thus, we can write,

$$P(A \rightarrow fixedServer) > P(A \rightarrow randomServer).$$

$\square$

### CONVERGENCE ANALYSIS

To analyze the convergence of the proposed QFL Algorithm 1, let's follow standard convergence analysis for SGD with the following assumptions.

**Assumption 1 (Smoothness)** *Loss function for each worker $L_i(x)$ is L-smooth i.e. $\forall (x, y) \in \mathbb{R}^d$,*

$$\mid L_i(x) - L_i(y) - \langle \Delta L_i(y), x - y \rangle \mid \leq \frac{L}{2} \parallel x - y \parallel^2 .$$

where, $\Delta L_i(y)$ is the gradient of $L$ at point $y$ and $\parallel x - y \parallel$ is the Euclidean distance between $x$ and $y$. Also, $L$ is the Lipschitz constant which is a mathematical concept to measure how fast a function can change. L-smooth loss function means its slope doesn't change too rapidly.

**Assumption 2 (Convex)** *The loss function for each worker $L_i(x)$ is $Convex$.*

In optimization, it is essential to find the global minimum and guarantee the convergence and efficiency of the algorithms.

**Assumption 3 (Unbiasedness and Bounded Variance)** *Gradients hold properties of unbiasedness and bounded variance.*

To analyze the convergence properties of the QFL algorithm, let us consider the following optimization problem:

$$\min_{\boldsymbol{w}} \frac{1}{n} \sum_{i=1}^{n} L_i(\boldsymbol{w}), \tag{1}$$

where $\boldsymbol{w}$ is the model parameter, $L_i(\boldsymbol{w})$ is the loss function for the $i$-th device, and $n$ is the total number of devices.

The stochastic gradient descent algorithm for QFL involves the following update rule:

$$\boldsymbol{w}_{t+1} = \boldsymbol{w}t - \eta_t \sum i = 1^n \nabla L_i(\boldsymbol{w}_t), \tag{2}$$

where $\eta_t$ is the learning rate at iteration $t$.

**Remark 1** *Standard SGD converges to a stationary point of the objective function with a rate of $O(1/\sqrt{T})$, where $T$ is the total number of iterations.*

Under the assumptions, we can show that SGD converges to a stationary point of the objective function with a rate of $O(1/\sqrt{T})$, where $T$ is the total number of iterations.

We show that the iterates $\boldsymbol{w}_t$ converge to a limit point, which is a stationary point of the objective function. We can use the following inequality to bound the expected distance between the iterates and the limit point:

$$\mathbb{E}[|\boldsymbol{w}_t - \boldsymbol{w}^*|^2] \leq \frac{2B^2}{\eta_t t} \tag{3}$$

where $\boldsymbol{w}^*$ is the limit point, $B$ is an upper bound on the norm of the gradients, and $\eta_t$ is the learning rate at iteration $t$.

Using this inequality, we can show that the expected objective function value converges to the optimal value at a rate of $O(1/\sqrt{T})$. Specifically, we have:

$$\mathbb{E}[f(\boldsymbol{w}_t)] - f(\boldsymbol{w}^*) \leq \frac{2B^2 L}{\sqrt{T}}, \tag{4}$$

where $f(\boldsymbol{w})$ is the objective function, and $L$ is the Lipschitz constant of the gradients.

Therefore, we have shown that SGD converges to a stationary point of the objective function with a rate of $O(1/\sqrt{T})$, under suitable assumptions on the loss functions, gradients, and learning rate.

CONVERGENCE CONDITIONS OF ALGORITHM 1

**Remark 2** *PQC scheme Dilithium has a time complexity of $O(1/\lambda^2)$ where $\lambda$ is the security level of the scheme.*

With security level $\lambda$, the key generation happens only once which takes time as $keyTime()$. For signing, the time it takes is $signTime()$ whereas, for verification, the time it takes will be $verificationTime()$. The performance of Dilithium depends on many factors like hardware implementation, message size, etc. Here,

$$\{keyTime() + signTime() + verificationTime()\} \propto \lambda$$

**Theorem 2** *Given $n$ devices, each device trains local stochastic gradient descent (SGD) with $T$ iterations and employs PQC at security level $\lambda$. Then Algorithm 1 converges with a rate of $n * \{O(1/\sqrt{T}) + O(1/\lambda^2)\}$.*

PROOF. From remark 1, we get the SGD convergence rate as: $O(1/\sqrt{T})$. Whereas with the added time complexity of the PQC scheme for signing, verifying, and key generating, the proposed scheme converges at the rate of $O(1/\sqrt{T}) + O(1/\lambda^2)$. Where PQC time complexity is $O(1/\lambda^2)$. Thus, the total time delay for convergence rate would be at least $\leq n * \{O(1/\sqrt{T}) + O(1/\lambda^2)\}$ for $n$ devices. $\square$

SECURITY CONDITIONS

For each communication between the trainer device and server device by the following Algorithm 1, the security level is SUF-CMA (Strong Existential Unforgeability under Chosen Message Attack) secure which is provided by the Dilithium signature scheme.

For any digital signature scheme, the standard notion is UF-CMA security which refers to security under chosen message attacks. The security model involves an adversary with an accessible public key that can be used to access the signing oracle to sign other messages. The adversary tries to get a valid signature for any new message. Dilithium signature scheme is SUF-CMA secure i.e.

Strong Unforgeability under Chosen Message Attacks. It is based on the hardness of standard lattice problems.

With Dilithium implementation, the communication between server and client becomes SUF-CMA secure i.e. for quantum random oracle H, adversary $\mathcal{A}$ has the advantage of breaking the communication which can be represented as (Ducas et al.):

$$Adv_{Dilithium}^{SUF-CMA}(\mathcal{A}) \leq Adv_{k,l,D}^{MLWE}(\mathcal{B}) + Adv_{H,k,l+1,\zeta}^{SelfTargetMSIS}(\mathcal{C}) + Adv_{k,l,\zeta'}^{MSIS}(\mathcal{D}) + 2^{-254} \quad (5)$$

for uniform distribution $D$ over $\mathcal{S}_n$. Also,

$$\zeta = max\{\gamma_1 - \beta, 2\gamma_2 + 1 + 2^{d-1} * 60\} \leq 4\gamma_2$$

$$\zeta' = max\{2(\gamma_1 - \beta), 4\gamma_2 + 2\} \leq 4\gamma_2 + 2$$

In Eqn. 5, the assumptions used are:

1. $Adv_{k,l,D}^{MLWE}(\mathcal{B})$ is an advantage of algorithm $B$ in solving MLWE (Module variant of Learning with Error) problem for integers $m, k$ with probability distribution $D : R_q \to [0, 1]$. This assumption is required to protect against key recovery.

2. $Adv_{H,k,l+1,\zeta}^{SelfTargetMSIS}(\mathcal{C})$ refers to advantage of algorithm $C$ in solving SelfTargetMSIS problem.It is based on the combined hardness of MSIS and the hash function $H$. Self-TargetMSIS assumption provides the basis for new message forgery.

3. MSIS assumption is needed for strong unforgeability.

The notations used are $k, l$ are integers, $D$ is a probability distribution, $\mathcal{A}, \mathcal{B}, \mathcal{C}, \mathcal{D}$ are algorithms, $H$ is a cryptographic hash function, $2^{-254}$ is a small constant representing negligible probability of occurrence of some rare event.

Dilithium is simple to implement securely, stateless, and acceptable combined size of public key and signature (Ducas et al.).

