# OpenReview forum: "SECURE COMMUNICATION MODEL FOR QUANTUM FEDERATED LEARNING: A PROOF OF CONCEPT"
_ICLR.cc/2023/TinyPapers — Submitted to Tiny Papers @ ICLR 2023_

### Official Review · Reviewer_VN6T · 2023-03-29

**Confidence:** 4

**Summary Of Contributions:**

A preliminary study is presented in this article with a proof of concept model for a post-quantum secured quantum federated learning system. The authors develop a novel PQC-QFL model. A proof of concept model is developed and a theoretical and experimental evaluation of the proposed model is conducted in this article.

**Rating:**

High Potential (HP): a submission which meets the reviewing criteria and has potential to make an impact on the field

**Strengths And Weaknesses:**

## Strengths
- The findings are communicated effectively via algorithms and figures.
- The results are reproducible with the provided source code links.
- The article follows the basic requirements of submission and follows code of conduct

## Weakness
- Author fails to introduce the Post Quantum Cryptography Quantum Federated Learning concepts in simple terms.
- Abstract misses clear picture of the article on which it is conducted.


**Suggested Changes:**

- Please communicate the implementation of your results clearly on the source code website.  Viewer misses if the article and provided source code link is for the same result you demonstrated.
- It would be great (not necessary) to show some examples in the appendix or external links.
- I got impression that article is overloaded with information, due to which reader misses the critical points.

---

### Official Review · Reviewer_KdLn · 2023-04-03

**Confidence:** 3

**Summary Of Contributions:**

This paper talks about a simple PoC framework to enhance the communication security of quantum federated learning. Given this framework, authors prove the convergence rate and resilience.

**Rating:**

Clear, Correct, and Reproducible (CCR): a submission which meets the reviewing criteria

**Strengths And Weaknesses:**

**Strengths:**

S1: The idea for enhancing communication security for quantum federated learning is novel. As far as I know, this is the first work try to tackle this problem.

S2. This paper is well written. The logic is easy to follow. All the findings in the paper is expressed clearly and effectively.

S3. Experiment setup is clear and easy to be reproduced. Experimental results on MNIST are strong enough to cover the statements in the paper (like little latency impact, training/test accuracy).

S4: The claims and conclusions are justified by the findings. Authors follow the formatting requirements.

**Weaknesses:**

W1: The notation in Algorithm 1 is not consistent. What is $p_d$ and $pk_d$? `fedAveraging` should be `FedAvg`.

W2: I assume the experiment is done by applying Dilithium to FedAvg. Could this approach be generalized to other federated learning approach? like fedprox, moon, FedAdam?

**Suggested Changes:**

C1: As stated in W1, W2, could this approach be easily used for other FL algorithms?

C2: For the related work section, authors should talk broadly about the general FL, not only QFL. (e.g. the following papers)

[1] Communication-Efficient Learning of Deep Networks from Decentralized Data

[2] Federated Optimization in Heterogeneous Networks

[3] Federated Multiple Label Hashing (FedMLH): Communication Efficient Federated Learning on Extreme Classification Tasks

[4] Adaptive Federated Optimization

---

### Meta-Review · Area_Chair_1VkA · 2023-04-04

**Recommendation:** Invite to present (notable)
**Confidence:** 3

**Metareview:**

The reviewers agree that the idea is novel and the paper is well written. Experiments are also strong enough. Claims and conclusions are well justified. The writing may be further improved.


**Summary:**

This paper studies the communication security of quantum federated learning, with a proved  convergence rate and resilience, as well as experiments.

**Reason For Not Giving A Higher Recommendation:**

The writing may be further improved.


**Reason For Not Giving A Lower Recommendation:**

The idea is novel and the paper is well written. Experiments are also strong enough. Claims and conclusions are well justified.

---

### Decision · Program_Chairs · 2023-04-07

**Decision:**

Invite to present (notable)

**Comment:**

Please add your URM statement.

---

> ### Comment · Area_Chair_1VkA · 2023-06-01
> **Archival**
>
> This work meets the threshold for archival, contents the URM statement and is deanonymized